# Adeno-Associated Virus Mediated Gene Therapy for Corneal Diseases

**DOI:** 10.3390/pharmaceutics12080767

**Published:** 2020-08-13

**Authors:** Prabhakar Bastola, Liujiang Song, Brian C. Gilger, Matthew L. Hirsch

**Affiliations:** 1Ophthalmology, University of North Carolina, Chapel Hill, NC 27599, USA; pbastola@email.unc.edu (P.B.); liujiang@email.unc.edu (L.S.); bgilger@ncsu.edu (B.C.G.); 2Gene Therapy Center, University of North Carolina, Chapel Hill, NC 27599, USA; 3Clinical Sciences, North Carolina State University, Raleigh, NC 27695, USA

**Keywords:** adeno-associated viruses, cornea, corneal diseases, AAV gene therapy, rAAV

## Abstract

According to the World Health Organization, corneal diseases are the fourth leading cause of blindness worldwide accounting for 5.1% of all ocular deficiencies. Current therapies for corneal diseases, which include eye drops, oral medications, corrective surgeries, and corneal transplantation are largely inadequate, have undesirable side effects including blindness, and can require life-long applications. Adeno-associated virus (AAV) mediated gene therapy is an optimistic strategy that involves the delivery of genetic material to target human diseases through gene augmentation, gene deletion, and/or gene editing. With two therapies already approved by the United States Food and Drug Administration and 200 ongoing clinical trials, recombinant AAV (rAAV) has emerged as the in vivo viral vector-of-choice to deliver genetic material to target human diseases. Likewise, the relative ease of applications through targeted delivery and its compartmental nature makes the cornea an enticing tissue for AAV mediated gene therapy applications. This current review seeks to summarize the development of AAV gene therapy, highlight preclinical efficacy studies, and discuss potential applications and challenges of this technology for targeting corneal diseases.

## 1. Introduction

Acquired or inherited variations in genomic DNA can lead to suboptimal, malfunctioning, or nonfunctional proteins, resulting in reduced cellular fitness and ultimately disease [1]. More than 75,000 disease-associated genetic variations have been reported to date [2], and most of these diseases currently lack effective therapies. Gene therapy is an optimistic strategy that seeks to deliver genetic material to targeted cells for the biological correction of genetic or acquired disorders. This approach has the potential to treat such disorders in multiple ways: (i) providing the normal copy of the gene (or cDNA) to the target cells to replace inactivating mutations (such as *F9* gene replacement for hemophilia B [3]), (ii) knocking down or knocking out genes with activating mutations (such as siRNA, shRNA, or designer endonuclease downregulation/knock out of *mHTT* (mutant huntingtin gene) in Huntington’s disease [4]), (iii) delivering genes associated with improving disease phenotypes (such as delivering VEGF inhibitor, sFlt-1, and to reduce retinal neovascularization [5]), or (iv) precise editing of underlying mutations at the genomic loci (such as editing pathogenic *CFTR* mutations in cystic fibrosis [6]). Once the necessary genetic locus is identified and depending on the disease context, gene therapies can be delivered to the target cells outside the body (ex vivo)—where target cells are harvested from the patient, genetically modified and subsequently infused back into the patient—or it could be delivered to the target cells inside the body (in vivo). To facilitate the delivery of such gene drugs, different non-viral and viral vehicles have been developed over the past decades. Common non-viral delivery approaches include electroporation, liposomes, and polymers [7]. Although non-viral approaches are considered less likely to induce an immune response, some non-viral approaches are not practical in clinical settings, while other approaches might result in minimal on-target efficiency rendering them currently less desirable in a gene therapy context (reviewed in [7,8]). Likewise, several viral vectors such as adenovirus, retrovirus, lentivirus (subtype of retrovirus), and adeno-associated virus (AAV) have been evaluated as delivery vehicles to treat metabolic, hematological, ophthalmological, muscular, infectious disorders, and cancers [9]. Currently, two lentiviral-based vectors are approved by the United States Food and Drug Administration (USFDA) for ex vivo Chimeric antigen receptor T-cell (CAR-T) therapy against some form of hematological malignancies [10], while two AAV-based gene therapies Luxturna^®^, and Zolgensma^®^, have been approved for in vivo gene therapy applications for RPE65-associated Leber’s congenital amaurosis [11], and spinal muscular atrophy [12], respectively. According to the NIH-Clinical Trials portal (https://clinicaltrials.gov/), as of 23 April 2020, there are 209 registered recombinant AAV (rAAV) human clinical trials targeting disorders such as hemophilia B, hemophilia A, Pompe disease, X-linked retinitis pigmentosa, and Parkinson’s disease. Further highlighting the promise of AAV vectors drug delivery, estimations by the USFDA have predicted that 10–20 new AAV gene therapy drugs will be approved by 2025. As such, AAV vectors, have emerged as the in vivo vector-of-choice due to its relatively low immunogenicity, broad tropism, and excellent safety profile.

AAV gene therapy platform has been pursued as potential therapy for several corneal diseases, which will be the focus of this review. According to the World Health Organization, corneal diseases remain the fourth leading cause of blindness globally after cataract, glaucoma, and age-related macular degeneration [13]. Corneal diseases can be caused by mechanical injuries, chemical burns, allergy, infections, or underlying genetics [14]. The clinical manifestation of corneal diseases can vary greatly, from very mild corneal discomfort to severe discomfort that ultimately results in blindness. Over the years, many treatment modalities including topical medications, enzyme replacement therapy, corneal surgery, and corneal transplantation have been developed for corneal diseases [14]. Although corneal transplantation is one of the most commonly performed transplantation surgeries in the world, this procedure is limited due to a lack of donor tissue, and high rates of rejection in some diseases. The clinical success of AAV based gene therapy in the retina highlights the potential of gene therapy towards designing successful therapies for the cornea. This review provides a brief overview of AAV-based gene therapy while highlighting potential AAV gene therapy approaches and challenges towards developing novel therapies for corneal diseases.

## 2. Adeno-Associated Virus (AAV) Background

First discovered in 1965 as a contaminant in simian adenovirus type 15 (SV15) preparations from rhesus monkey kidney cell cultures infected with SV15 [15], AAV is classified under the genus *Dependovirus*, which belongs to the Parvoviridae family. Initially shown to replicate upon adenovirus infection [15], subsequent studies indicated that AAV replication can be achieved through coinfection of other helper viruses, such as herpesvirus [16] and human papillomavirus [17], or in the absence of coinfection by cellular stress [18,19,20]. AAV spans approximately 25 nm in diameter, making it among the smallest viruses identified to date [21]. This nonenveloped virus is composed of three outer icosahedral capsid proteins (VP1, VP2, and VP3), which harbor a single-stranded DNA genome of 4.7 kb. AAV genome primarily includes two main protein coding regions and one poly-A tail, flanked at both ends by 145 nucleotides inverted terminal repeats (ITRs). The two main protein coding regions, *rep* and *cap*, code for at least nine different proteins including four Rep protein variants, three capsid protein variants, the assembly activation protein (AAP) and the Membrane-Associated Accessory Protein (MAAP) [22,23,24].

The AAV genome is flanked by the ITRs, which play important roles in the viral life cycle such as, replication, encapsidation, episomal persistence, and integration. The ITRs are comprised of 145 nucleotide sequences residing at each end of the AAV genome [25]. These sequences form palindromic double-stranded T-shaped hairpin-structures through self-annealing, and harbor Rep binding element (RBE) and terminal resolution site (TRS) sites that are necessary for viral replication [26]. Several studies have indicated that the Rep proteins (Rep78/68) remain attached to the nascent AAV genome and subsequently dock on the five-fold (5F) axis of the newly assembled capsid [27,28,29]. The AAV genome is thought to be inserted into the capsid via the helicase activity of Rep52/40 proteins [30]. After viral infection, AAV genomes are reported to persist primarily as extrachromosomal episomes as head-to-tail circular concatemers; however, a small fraction of the AAV genome can integrate into a specific location on human chromosome 19, known as the AAVS1 locus, which contains a tetrad repeat very similar to RBE [21]. Multiple AAV genomes can recombine at the ITR sites to create episomes inside the host nucleus. These episomes can persist in dividing as well as non-dividing cells. In non-dividing cells, AAV have been shown to transcribe for up to 12 years [31]; whereas in dividing cells, episomes can get diluted in the absence of a helper virus due to limited AAV replication. In the absence of a helper virus, the AAV genome is thought to establish latency either as episomes and/or via site-specific genome integration within the host cell genome. Integration is preferred in dividing cells, and this process is mediated through the interaction between the Rep68/78 proteins and ITRs at the AAVS1 locus, which harbors sequence similarity with the ITRs. Upon coinfection of cells with helper viruses, AAV enters the lytic stage where AAV genome replication gets initiated through transcriptional activation via the alleviation of the Rep mediated repression at the p5 promoter region. In addition, helper viruses have been shown to aid in the AAV viral life cycle by promoting the second-strand synthesis of AAV, inhibiting p53-mediated apoptosis, down-regulating host cell translation, promoting AAV mRNA transport to the cytoplasm, and attenuating cells in S-phase [32]. Active replication allows for the packaging of AAV particles followed by the release of these particles, and subsequent infection of additional cells. In summary, the AAV particle harbors a relatively small yet remarkably complex genome, and through coinfection of helper viruses into the host cells, AAV is capable of expressing its genes, persisting, and/or integrating into the host genome, replicating, and packaging its genome to ultimately infecting additional cells.

## 3. Developing AAV as a Vector for In Vivo Gene Delivery

AAV harbors several properties that renders vectorization ideal for gene delivery in vivo. Although, multiple serotypes of AAV have been isolated from human and animal tissues (at least 12 naturally occurring serotypes and more than 100 variants), no AAV serotype has been associated with any known disease in these organisms [33,34]. AAV serotypes display a broad host tissue tropism [35] and can infect dividing and non-dividing cells, which is an advantage for systemic gene delivery [36]. In addition, the different AAV serotypes display preferential transduction efficiency towards certain cell type, which is idyllic for cell specific targeted gene delivery in vivo [35]. AAVs generate a relatively mild innate and adaptive immune response [37], in part, allowing for sustained expression of the therapeutic gene. Overall, these properties rationalized the efforts to develop AAV vectors for gene therapy uses during the 1980s.

In 1982, Samulski et al. reported the cloning of the intact AAV2 genome into a bacterial plasmid. Upon coinfection with the adenovirus, the group showed that this new AAV2 plasmid replicated and produced infectious AAV2 virions indistinguishable from the original wild-type virus [38]. Subsequently, Hermonat et al. (in 1984) transfected a neomycin resistance gene inserted into an AAV truncated plasmid (harboring cap deletion) and an enlarged AAV plasmid harboring *rep*, *cap,* and a nonessential region into Ad2 infected human cells to show that foreign DNA could be replicated and packaged using the AAV genome [39]. Simultaneously, Tratschin et al. (in 1984) also showed that they could replicate and package a novel rAAV deletion plasmid harboring chloramphenicol acetyltransferase (CAT) by supplying plasmid expressing *rep* and *cap* together with adenovirus infection [40]. In both of these reports, the newly packaged rAAV particles could subsequently infect cells to confer resistance against the inserted antibiotic resistance gene. These reports (Hermonat et al. and Tratschin et al.) clearly showed that AAV vectors could be engineered to transduce and express foreign DNA in human cells; however, these studies resulted in the undesired production of wild-type AAV2 and adenovirus as contaminants. Later, Samulski et al. (in 1989) showed replication, packaging, and subsequent infection of the neomycin marker containing AAV plasmid flanked with the 191 nucleotide, which included the AAV2 ITRs, while providing the AAV *rep* and *cap* ORFs through a separate helper plasmid flanked by adenovirus terminal repeats in the presence of an adenovirus [41]. This report demonstrated that the new AAV ORF helper plasmid (void of the AAV terminal sequences) was not replicated and therefore resulted in greatly reduced wild-type AAV contamination when packaging the rAAV virions. Finally, Xiao et al. (in 1998) created a defective adenovirus plasmid incapable of generating Ad particles, but capable of yielding rAAV virions when transfected together with rAAV plasmid and a plasmid expressing AAV *rep* and *cap* [42]. Therefore, by nullifying the production of Ad particles post transfection, this approach allowed for the enrichment and purification of rAAV particles during the preparation. These seminal studies laid the foundation for AAV vector design and production for gene delivery applications. Current AAV production still uses most of these components and protocols with some minor modifications.

## 4. Targeting Disease with AAV Gene Therapy

AAV has emerged as a promising platform to deliver genetic material in vivo. To highlight key steps involved prior to submitting the IND (investigational new drug) application to the USFDA using AAV based gene therapy, studies performed to target the corneal disease Mucopolysaccharidoses I (MPS I) will be discussed below as an example; however, the overarching concepts for AAV gene therapy development remains largely similar across different diseases. MPS I is a progressively debilitating monogenetic metabolic storage disease caused by loss-of-function mutations in both copies of the *IDUA* gene. MPS I patients display progressive corneal clouding, which can result in impaired vision; however, current treatments such as hematopoietic stem cell transplantation and systemic enzyme replacement therapy fail to alter the corneal clouding phenotype. MPS I has been characterized as a disease of the corneal stroma (middle layer of the cornea, discussed in Section 5) [43,44]; therefore, it was hypothesized that the delivery of recombinant *IDUA* in the corneal stroma could resolve the corneal clouding phenotype in these patients. Corneal stroma harbors quiescent corneal fibroblast cells or keratocytes, which are ideal for AAV based gene therapy as AAV episomal DNA would not be diluted upon cell division. Once the target cells were identified, subsequent studies focused on optimizing the gene delivery to the target tissue [45]. This included identifying AAV capsids that could transduce corneal keratocytes and optimizing the delivery route to target corneal keratocytes in vivo. Towards this realm, AAV8 [46] and AAV9 [47] were previously deemed most efficient for corneal transduction. Therefore, an AAV8 capsid scaffold engrafted with the AAV9 putative galactose binding domain was engineered and evaluated in human corneas recovered post-mortem following intrastromal injection. In this context, the AAV8/9 chimeric capsid (8G9) resulted in greater transduction compared to either parent serotypes at equivalent vector doses [45]. Therefore, capsid AAV8G9 was chosen as the capsid of choice for further clinical development. The next step in the development of an AAV based gene addition therapeutics was the synthesis of the optimized AAV therapeutic cassette, which incorporated the *IDUA* cDNA (2.2 Kb) to supplement the defective native locus in MPS I patients along with the regulatory elements—minimally a promoter, a Kozak sequence, and a polyadenylation signal (Figure 1). The *IDUA* cDNA was optimized to eliminate alternate open reading frames and to decrease regions of significant DNA secondary structures that have been reported to affect capsid packaging integrity [48]. As all cells express the *IDUA* protein, ubiquitous promoters were chosen for expression, originally CMV [45], however, due to upregulation during cellular stress, it was later replaced with the human EF1α [49]. It is important to note that depending on the therapeutic cDNA and native expression a cell restricted promoter may be desirable based on the disease application to preferentially express the transgene in the tissue-of-interest and thereby reduce deleterious effects due to off-target expression. Depending on the overall size of the protein coding region and the regulatory elements, ITRs can be chosen to a package single stranded AAV genome or a self-complementary AAV genome, which is enhanced for the overall transgene expression. The MPS I example used in Figure 1 harbors a ubiquitous EF1α promoter, codon optimized *IDUA,* and SV40 poly adenylation sequence totaling 3.3 Kb in size, necessitating the use of the wild-type ITR sequence resulting in the packaging of single strand rAAV genomes. However, if the size of the overall protein coding sequence and the regulatory elements fell below 2.2 Kb, a deleted ITR could be included in the AAV plasmid to create a self-complementary genome, which not only circumvents the need for second strand synthesis, which is a rate limiting step in the AAV transduction [50], but also allows for faster and increased transgene expression [51,52].

Now, to replicate and package the opt-IDUA (optimized-IDUA) cassette, a typical protocol involved transfection of three plasmids into the HEK293 cell line, including (1) the WT ITR-flanked opt-IDUA, (2) the plasmid expressing AAV *rep2* and *cap8G9* (with no ITRs), and (3) a plasmid encoding Ad helper functions to the promoter overall production [53] (Figure 1). This approach allowed rAAV genomes encoding the gene expression cassette to replicate and then be packaged inside the AAV8G9 capsids, which was subsequently purified by various methods [53,54]. The purified rAAV virions then need to be titered, which is most often performed by Q-PCR, ddPCR, and/or Southern blotting techniques that detect DNAse resistant (or encapsidated) vector genomes—the unit of measurement for rAAV. Additionally, alkaline gel electrophoresis of packaged genomes informs of packaged genome integrity and the overall titer. Once the purified virions were obtained, studies were conducted to show rAAV8G9-opt-IDUA viral transduction in ex vivo human keratocytes [45], and relevant animal cornea models such as rabbits [55] and MPS I canines [49]. It is important to note that rAAV viral transduction is a multi-step process, which include viral binding to the cell surface, viral entry, endosomal trafficking, endosomal escape, nuclear import, uncoating, and second strand synthesis (reviewed in [49,56,57]). The mechanisms and cellular components involved in rAAV vector transduction are not fully understood; therefore, further elucidation of these mechanisms have the potential to enhance viral transduction in vivo. Lastly, *IDUA* gene therapy resulted in clearing of corneal clouding phenotype as well as no apparent toxicity was observed in the treated animals [45,49]. Additionally, corneal gene therapy has perhaps the unique ability to not only investigate gene delivery in the intended target tissue of the desired species, it also allows for relatively short-term toxicities studies. This is of particular importance as AAV drug development in animal species is becoming increasingly appreciated to lend translational concerns when tested in human patients. To summarize, a simple recipe for the development of AAV gene therapy pre-IND application include identifying the disease, optimizing the gene delivery to the target tissue/cells, optimizing the therapeutic cassette, and testing in relevant animal models to show efficacy as well as minimal toxicity.

## 5. Cornea: Structure and Function

Light first enters the eye through a nearly transparent, avascular, tough, and flexible tissue called the cornea. Along with the sclera, the cornea comprises of the outermost tissue of the eye, and serves as a structural barrier to environmental insults such as infection and debris. Likewise, the cornea accounts for two-thirds of the refractive power in the eye, highlighting its crucial role in vision [58]. Structurally, the cornea can be subdivided into three cellular layers: the outermost epithelium, the intermediate stroma, and the innermost endothelium. There are also two acellular layers: Bowman’s layer and Descemet’s membrane [59] (Figure 2). The corneal epithelium is composed of five or six layers (50 µm thick) of non-keratinized squamous epithelium [60]. These cells serve as the barrier between the external environment and the stroma. Additionally, the outermost corneal surface cells aid in the adhesion of the tear film via microvilli and microplicae structures, and by secreting the adhesive enzyme glycocalyx. Stresses such as mechanical friction associated with constant eye blinking and ultraviolet radiation shed the outer epithelial layer regularly, which is replaced by replicating cells of the underlying basal epithelial layer. The human corneal stroma, which is 500 µm in size, makes up the vast majority (90%) of the corneal thickness. It contains a sparsely distributed population (2–3% by volume) of quiescent corneal fibroblast cells or keratocytes, while the remaining stromal extracellular matrix is composed of intricately stacked acellular components including collagen fibrils, elastins, fibronectins, and proteoglycans. During embryonic development, keratocytes synthesize and secrete collagen types including I, III, V, VI, XII, and XIV and other components necessary to create the stromal extracellular matrix (reviewed in [61]). Bundles of collagen fibrils referred to as lamella are intricately arranged to provide physical strength, stability of shape, and transparency. Finally, the endothelium is comprised of corneal endothelial cells, which form a monolayer (5 µm thick) and acts as a barrier between the aqueous humor and the stroma [62]. Corneal endothelial cells maintain corneal homeostasis by allowing the passage of nutrients and removal of metabolic waste to and from the stromal cells. Additionally, corneal endothelial cells are equipped with enzymatic pumps that allow the maintenance of an ionic gradient between the cornea and aqueous humor resulting in the extraction of water from the stroma, which is essential for corneal transparency [62]. Lastly, the human cornea also displays two acellular membranes, a Bowman’s layer located between the epithelium and stroma and the Descemet’s membrane located between the stroma and the endothelium, which provide structural support, elasticity, and an attachment matrix for the subsequent layers.

The corneal surface is exposed to the environment; therefore, it relies on a diverse array of defense mechanisms to prevent pathogen invasion while simultaneously tolerating normal ocular surface microbial flora. To prevent microbial invasion, the cornea harbors surface mucins [63] and epithelial barriers, antimicrobial tear film proteins, antigen-recognition receptors called Toll-like receptors in the epithelium, and a reactive lacrimal gland that liberates white blood cells and IgA in response to injury [64]. At the same time, the normal cornea is considered a relatively immune privileged site that is tolerant of specific antigens. This immunological tolerance is maintained by multiple factors, including the blood-ocular barrier, a lack of corneal lymphatics and vasculature, the expression of Fas ligand on corneal cells, low expression of major histocompatibility complex (MHC) class I and II molecules on corneal cells, few mature antigen presenting cells (APCs) in the central cornea, the presence of immunomodulatory factors in the aqueous humor and tear film (e.g., alpha melanocyte stimulating hormone and transforming growth factor beta), and the phenomenon of anterior chamber associated immune deviation (ACAID), where down regulation of systemic delayed-type hypersensitivity results from the introduction of alloantigens to the anterior chamber [65]. Breakdown of any of these factors may lead to infectious keratitis; conversely, the development of an overwhelming inflammatory response may damage normal tissue resulting in blindness. Exposure to the environment also renders the corneal tissue more vulnerable to physical trauma, ultraviolet radiation, and chemical exposure. Lastly, underlying genetics can result in corneal abnormalities, collectively known as corneal dystrophies, which may cause significant vision impairment. In the subsequent section, we classify corneal diseases and discuss molecular mechanisms relevant to gene therapy.

## 6. Corneal Disorders

The National Eye Institute (NEI) breaks down corneal disorders into five conditions: injuries, allergies, keratitis, dry eye disease, and corneal dystrophies. Injuries include mechanical abrasions or other physical injuries to the cornea. Based on the nature of such injuries these conditions can either self-heal or may need medical attention in the case of corneal scarring and vision problems. Corneal allergies can be caused by irritants such as pollen, dust, mold, or pet dander. These conditions can often be controlled by irritant avoidance, eye drops, antihistamines, or allergy shots. Keratitis encompasses all inflammatory conditions linked to microbes—such as viruses, bacteria, fungi, or amebae. Infections associated with improper handling and storage of contact lens or due to viral infection from herpes simplex virus-1 (HSV-1) are some commonly observed forms of corneal keratitis. Keratitis can be treated with topical antibiotics, antifungal, or antiviral agents; however, severe inflammation, which can cause blindness may require a corneal transplantation. Dry eye is a multifactorial ocular surface condition, which is characterized by the loss of tear film homeostasis. According to the Tear Film and Ocular Surface Society (TFOS), dry eye is accompanied by ocular symptoms, which include tear film instability and hyperosmolarity, ocular surface inflammation and damage, and neurosensory abnormalities [66]. Dry eye is one of the most common ocular conditions with an estimated 16.4 million adults (≥18 years) diagnosed in the US making up 33% of the patient pool in eye care clinics [67]. Over-the-counter artificial tears, tear duct plugs, and prescriptions such as cyclosporine or lifitegrast have been used to combat dry eye symptoms; however, these treatments require 4–6 applications of the solutions per day, and thus patient compliance is a major issue; not to mention, these treatments are still not curative. Lastly, the corneal dystrophies include a group of noninflammatory slowly progressing inherited diseases that can lead to corneal deposits and morphological changes. To date, more than 25 types of corneal dystrophies have been identified (https://disorders.eyes.arizona.edu/), each with a different set of symptoms; however, they all cause a characteristic buildup of foreign deposits in the cornea. Corneal dystrophies are rare with a combined prevalence rate of 897 per million in the United States [68]. Although mutations in genes such as *KRT3*, *KRT12*, *PIP5K3*, *TGFBI,* and *UBIAD1* have been associated with specific corneal dystrophies, genes associated with all corneal dystrophies have yet to be identified. Such studies will help to elucidate the underlying disease pathologies of corneal dystrophies and aid in the development of therapeutics. Many of these corneal diseases may benefit from gene therapy, specifically AAV-mediated gene therapy. In the subsequent subsections, we will review corneal disorders primarily focusing on disorders that could be targeted by AAV-mediated gene therapy.

### 6.1. Mechanical Injuries and Chemical Burns

Being the anterior-most tissue structures of the eye renders the cornea vulnerable to various external hazards such as material debris, blunt force trauma, and chemical exposure. Due to the rapid onset of such injuries, a swift clinical response is tailored towards alleviating the symptoms, preventing future complications, and aiding in corneal healing [14]. Treatment modalities for mechanical injuries due to material debris may include topical cycloplegic agents, topical antibiotics, and/or application of a tight patch. Minor mechanical injuries tend to heal spontaneously; however, deep tissue penetration into the stroma increases the risk of infection. Subsequently, stroma injury may result in abnormal scar formation, which can result in alteration of the corneal surface and ocular abnormalities. Injuries from blunt force trauma can result in the rapid transfer of force, which tend to injure the epithelium and the endothelium. Significant blunt force trauma can induce subsequent inflammatory response; therefore, treatment is tailored towards reducing inflammation and lowering further damage to the ocular tissues. Lastly, corneal exposure to chemical products, which include acids, bases (mostly alkali), oxidizing/reducing agents, and corrosives can damage the corneal tissue. Immediate irrigation of the eye to remove chemical debris and subsequent mitigation of symptoms is considered to be the best management practices following chemical burns.

Following early interventions, subsequent treatment to restore or manage visual abnormalities include limbal stem cell transplantation, amniotic membrane grafts, and keratoprosthesis; however, such efforts may be inadequate. Corneal injuries can result in increased corneal neovascularization as well as increased release of cytokines, including transforming growth factor β (TGFβ) in the cornea that can induce the inflammatory response. Although the inflammatory response is important during the process of wound healing, excessive induction may result in corneal fibrosis and the loss of vision. Therefore, gene therapy tailored to reduce corneal neovascularization and corneal inflammation has been shown to improve clinical outcomes in animal models. AAV5-mediated targeted gene therapy of decorin, an inhibitor of TGFβ, has shown decreased corneal haze [69] and corneal neovascularization [70] in an in vivo rabbit model of corneal fibrosis. Likewise, topical application of AAV5-Smad7 (a negative regulator of TGFβ signaling) demonstrated a reduction in corneal hazing and corneal fibrosis, and a lack of immune cell infiltration in an in vivo rabbit model [71]. Adenovirus-mediated gene therapy has been used to express PPARγ, another negative regulator of TGFβ signaling, in keratocytes in a mouse alkali burn model showed encouraging epithelial layer recovery and reconstruction of the basement membrane [72]; however, the use of AAV-PPARγ has not been reported. Another strategy involved chimeric self-complementary AAV8G9 (scAAV8G9) mediated gene delivery of HLA-G, an immunomodulatory, and an anti-inflammatory molecule, following corneal alkali burn in rabbit model [73]. The study demonstrated significant reduction in cornea neovascularization and maintenance of immune cell infiltration of the cornea [73]. Lastly, gene therapy targeting corneal neovascularization post corneal damage might be beneficial in managing long term visual abnormalities—this will be discussed in further detail in Section 6.8 below.

### 6.2. Infectious Keratitis

Keratitis can be broadly divided into two main types: infectious keratitis and non-infectious keratitis. Infectious keratitis is caused by viruses, fungi, parasites, or bacteria (i.e., microbial keratitis) of which bacterial infections are most common. It is estimated to occur in 1.5–2 million people/year in Asia and Africa alone, where it is the most frequent cause of corneal blindness. Factors for developing microbial keratitis include improper contact lens use, dry eye, ocular trauma, or a mechanical or neurologic eyelid abnormality [14]. Treatment for these infectious diseases is usually based on a combination of systemic and topical antibiotics, topical corticosteroids, and antivirals, which are usually effective in controlling the active infections [74,75]. However, gene therapy could be used to prevent infection or to manage side effects, such as corneal fibrosis or haze [76], which commonly develops after infection and inflammation resulting from the disruption of the corneal stromal collagen or alterations in the corneal glycosaminoglycans. The potential of gene therapy for the prevention of stromal fibrosis was demonstrated using AAV5-mediated decorin gene therapy [69] and AAV-mediated HLA-G gene therapy [73]—discussed earlier in the Section 6.1.

Herpes simplex virus (HSV)-mediated keratitis is the most common cause of blindness due to infection observed in the developed world. It is estimated that 500,000 people in the United States have ocular disease associated with HSV [77], with 50,000 new and recurring cases diagnosed per year. Unlike other infectious keratitis (fungal keratitis and bacterial keratitis), HSV keratitis has the potential to become chronic and can recur often due to triggers such as stress, hormonal imbalance, trauma, and UV radiation [78]. Current antiviral regimens used to treat HSV-mediated keratitis primarily act on active disease by targeting the HSV DNA polymerase; however, they fail to clear the latent infection (reviewed in [79]). AAV gene therapy has been employed to target the latency associated transcript (LAT) region, the region in the HSV genome that expresses genes during latency. Delivery of scAAV2-LAT-targeting ribozyme in the eyes of rabbits with latent HSV-1 infection blocked viral reactivation in more than 60% of the infected animals [80]. Furthermore, in recent years, multiple studies have demonstrated that the lytic and quiescent HSV genome can be targeted using the CRISPR-Cas9 system in vitro [81,82,83]. It will be interesting to see if these effects can be recapitulated through in vivo studies.

In addition, exposure to infectious agents may have an immune-mediated, or autoimmune component to their pathogenesis and AAV gene therapy might be beneficial in the management of these complications. For example, corneal scarring and vascularization may develop as a result of chronic immunologic reactions provoked by HSV antigens [84]. AAV-gene therapy to inhibit vascularization has been reported in experimental animals [5,85,86] and therefore may be beneficial for prevention or treatment of chronic HSV or other immune-mediated keratitis. Intravitreal injection of AAV serotype 2 vectors expressing the vascular endothelial growth factor (VEGF) inhibitor sFlt-1 have been shown to reduce retinal neovascularization in mouse models [85,87]. Likewise, subconjunctival injection of rAAV vectors carrying recombinant endostatin (serotype 2) [5] and angiostatin [86] reduced corneal neovascularization in mouse models. Lastly, intrastromal injection of AAV mediated delivery of HLA-G (serotype AAV8G9), an immunomodulatory and anti-inflammatory molecule, prevented corneal neovascularization in rabbit models. rAAV targeted neovascularization will be discussed in Section 6.8 in this review.

### 6.3. Dry Eye Disease

Dry eye disease is a chronic, multifactorial disease that results in dysfunction of the lacrimal functional unit and is frequently associated with Sjögren’s syndrome and mechanisms of autoimmunity [88,89]. Dry eye is characterized by ocular discomfort and inflammation, and chronic dry eye causes decreased corneal epithelial thickness, scarring, vascularization, and vision impairment [90]. It is estimated that 6.8% of the adults in the United States have been diagnosed with some form of dry eye disease, with higher prevalence (8.8%) in women [67]. Broadly, dry eye etiologies can be subdivided into aqueous deficient dry eye, evaporative dry eye, or a combination of both [66]. A particularly severe form of the aqueous deficient dry eye is associated with chronic graft vs. host disease (GVHD) that develops from chronic inflammation resulting in impaired function of the lacrimal glands and associated blepharitis, conjunctivitis, and keratitis [91]. Dry eye disease is managed in most patients through antigen avoidance and symptomatic therapy such as artificial tears, antihistamines, and anti-inflammatory medications [92]; however, these therapies can be often inadequate in managing the underlying symptoms or have side-effects such as eye irritation, eye pain, or allergic reactions. More importantly, none of the current therapies for dry eye disease are curative.

Several studies have shown that dry eye disease can be targeted with gene therapy. Zhu et al. showed that adenovirus-mediated transduction of the viral IL-10 transgene in the lacrimal gland resulted in suppression of Sjögren’s syndrome-like features such as reduced tear production, accelerated tear breakup, ocular surface disease, and immunopathologic response in rabbit [93]. Likewise, Trousdale et al. showed that transient expression of an adenovirus vector containing a TNF inhibitor (AdTNFRIp55-Ig) in the lacrimal tissue of the established autoimmune dacryoadenitis rabbit model resulted in favorable clinical impacts and reduced pathological features [94]. Similarly, beneficial effects were observed in animal models receiving adenovirus vectors expressing erythropoietin (Epo) [95] and AAV2-mediated expression of aquaporin 1 (AQP1) [96] through submandibular duct cannulation. Given that anti-inflammatory medications such as cyclosporine and lifitegrast are currently used to manage dry eye disease in clinics, AAV gene therapy with transgenes targeting the immune response such as HLA-G, and TGFβ may also be beneficial for the treatment of these ocular surface diseases [73,86].

### 6.4. Corneal Dystrophies

Corneal dystrophies are heritable, bilateral (affecting both eyes), and often progressive disorders of the cornea that have been shown to affect corneal transparency and/or corneal shape and thus impair vision. Advances in genetic analyses have allowed for the identification of the causative genetic mutations associated with most corneal dystrophies. To date, more than 25 disorders have been characterized as corneal dystrophies. Although the prevalence of each disorder is relatively rare, collectively corneal dystrophies have an estimated prevalence rate of 897 per million in the United States [68]. Fuchs endothelial corneal dystrophy (FECD), the most common form of posterior corneal dystrophy, will be discussed in detail in Section 6.5. Current treatment modalities for corneal dystrophy vary substantially depending on the severity. In some instances, the corneal dystrophy can be asymptomatic and therefore require no treatment, while in other instances visual obscurity including complete loss of vision is observed, hence treatment modalities must be tailored and range from ointments and therapeutic contact lenses for mild symptoms to phototherapeutic keratectomy if the disorder progressively deteriorates over-time. If sight further declines following multiple rounds of phototherapeutic keratectomies, corneal transplantation is required. Corneal transplantation surgeries bring their own set of challenges, including a shortage of donor corneas and graft rejection. Therefore, by addressing the underlying genetic abnormality, gene therapy presents an optimistic opportunity towards the development of better therapies against corneal dystrophies.

With the advancements of molecular studies, corneal dystrophies that were previously characterized based on the affected corneal tissue were later classified based on the causative genetic mutation. According to the University of Arizona database of hereditary ocular disease (https://disorders.eyes.arizona.edu/), as of 17 May 2020, there are 29 disorders identified under corneal dystrophies with 22 disorders that have been associated with genetic mutations (Table 1). The monogenetic nature and Mendelian pattern of inheritance exhibited by most of these disorders make them attractive targets for gene therapy approaches. Among all the listed corneal dystrophies, 3 are classified as recessive, 23 are characterized as autosomal dominant, and 2 disorders are thought to be X-linked dominant (Table 1). Corneal dystrophies associated with recessive pattern of inheritance include, Corneal hereditary endothelial dystrophy 2 (CHED2), Corneal Dystrophy Gelatinous Drop-Like (CDGDL), and Corneal Dystrophy Macular with mutations in *SLC4A11*, *M1S1,* and *CHST6*, respectively (Table 1). Mutations in both copies of these genes create a loss-of-function phenotype; therefore, gene complementation strategies appear ideal to treat such disorders. In fact, the ORFs of these three genes ranges from 0.9 to 2.7 kb making them ideal targets for an AAV-based gene complementation approach. For targeting autosomal dominant corneal dystrophies, a gene editing approach that can correct or knock-out the gain of toxic function mutant gene copy at the genomic locus may be more effective. To achieve this, programmable nucleases such as: meganucleases, zinc finger nucleases (ZFNs), transcription activator-like effector nucleases (TALENs), and clustered regularly interspersed palindromic repeats (CRISPR)/Cas nucleases have been successfully employed to target mammalian cells in vivo (reviewed in [97,98]). Among the above-mentioned nucleases, CRISPR-Cas based genome editing has gained popularity in recent years due to its relatively simple design for targeting genomic loci. While modifications in the DNA-binding protein domains are required to target with ZFNs and TALENs, a complementary spacer sequence (usually 20 nucleotide in length) in the single guide RNA (sgRNA) upstream of the protospacer adjacent motif (PAM) is sufficient to target the CRISPR-Cas9 endonuclease to the genomic locus of interest [99,100]. Binding of the Cas9 nuclease at the genomic locus results in a double strand break, which can be repaired either via error prone non-homologous end joining (NHEJ), or homologous directed repair (HDR) when a repair template is provided. NHEJ can create insertions and deletions in the gene resulting in a premature stop codon or disruption of the mutant ORF, which can knock-out the expression of the targeted protein. Moreover, HDR allows for the incorporation of the specific alterations provided by the repair template. CRISPR-Cas nucleases have been further modified to improve single base editing (base-editing) [101] or to perform precise small insertions and deletions (prime-editing) [102]. Such approaches are elegantly described in the following reviews [103,104]. Lastly, single-stranded oligodeoxyribonucleotides (ssODNs) can introduce site-specific genome editing [105], albeit at lower efficiency.

The CRISPR-Cas system has been deployed in corneal dystrophies. Courtney et al. reported the targeting of PAM generated by the missense mutation in *KRT12* gene that caused Meesmann’s corneal dystrophy (MECD), which allowed for selective disruption of the mutant copy in the humanized MECD mouse model [106]. A similar approach showed successful disruption of *TGFBI* (transforming growth factor–β-induced) mutant alleles relevant for corneal dystrophies; however, cleavage of the wild-type copy was also observed [107]. Furthermore, HDR-mediated gene editing approaches have been used in corneal dystrophies. Taketani et al. showed correction of R124H mutation in *TGFBI*, which causes granular corneal dystrophy (GCD) in primary human corneal keratocytes in vitro [108]; however, HDR efficacy of corneal keratocytes in vivo was not evaluated by this group. AAV platforms have also previously been utilized to perform HDR-mediated repair of therapeutically relevant disease targets. One such approach includes the coinfection of dual AAV vectors, where the AAV-SaCas9 (size 3.1 Kb) is provided by one vector, while the sgRNA and the repair template is provided in the second AAV vector. This has been successfully implemented by using AAV8 to edit liver cells in mice in vivo [109]. CRISPR-Cas mediated HDR repair holds great promise in targeting monogenic dominant corneal dystrophies; therefore, studies should be conducted towards editing mutant alleles associated with corneal dystrophies.

### 6.5. Fuchs Endothelial Corneal Dystrophy

FECD is the most common form of posterior corneal dystrophy, characterized by the loss of corneal endothelial cells, disruption of the endothelial barrier, and a thickened and irregular Descemet’s membrane. These changes result in progressive corneal edema, corneal opacity, and decreased vision [110]. FECD can be early onset (first decade of life; rare) or late onset (starts second-third decade of life; progressive) [110]. The late onset FECD is approximately three times more frequently observed in females compared to males [111]. Clinical features of FECD include a decrease in corneal endothelial cell density, irregular corneal endothelial cell morphology (pleomorphism), and variation in corneal endothelial cell size (polymegathism). An additional clinical feature commonly includes the presence of corneal guttae (an extracellular matrix outgrowth on or within the Descemet membrane). FECD has been associated with an autosomal dominant inheritance; however, variable penetrance as well as multiple associated genes make the determination of inheritance challenging. The early form of FECD is commonly associated with mutations in *COL8A2*, while the late-onset FECD has been associated with mutations in *ZEB1*, *SLC4A11*, *TCF4,* and others (reviewed in [111]). There is currently no cure for FECD. Palliative treatments such as the topical use of hypertonic saline, anterior stromal puncture, amnion membrane grafts, or therapeutic contact lenses may improve clinical signs [112]; however if the patient symptoms progressively deteriorates, surgical options such as penetrating keratoplasty or endothelial keratoplasty (preferred) may be necessary. Although endothelial keratoplastic surgeries are highly effective, the global shortage of human cornea has warranted a considerable interest in the development of new therapeutics for FECD including Rho kinase (ROCK) inhibitors (reviewed in [113]).

An AAV platform can be used to target FECD through multiple avenues. In FECD associated with *SLC4A11* mutations, which can result in *SLC4A11* misfolding and retention in the endoplasmic reticulum (ER), an AAV gene therapy approach focused on increasing the cell surface localization of *SLC4A11* may be beneficial therapeutically [114]. Additionally, studies have now delineated that trinucleotide intronic repeat expansion of CTG nucleotide in the *TCF4* transcription factor results in widespread changes in mRNA splicing [115]. Therefore, an AAV mediated CRISPR-Cas approach could be employed to target the aberrant *TCF4.* Lastly, in FECD treated with corneal grafting, gene therapy to prevent corneal graft rejection, as discussed in Section 6.7, or gene therapy to reduce fibrosis may improve treatment outcomes and vision [82].

### 6.6. Corneal Opacity Associated with Mucopolysaccharidoses

Mucopolysaccharidoses (MPS) represents a group of inherited disorders caused by defects in lysosomal enzymes required for the degradation of glycosaminoglycans (GAGs). With an estimated incidence of approximately 1:25,000 live births [116], patients with these disorders can manifest symptoms in the face, nervous system, ear, heart, bone, and cornea. Although corneal manifestations occur in almost all types of MPS (MPS I—IX), such manifestations are a common occurrence in MPS I, VI, and VII. Bone marrow transplantation and enzyme replacement therapy may improve the majority of the patient’s clinical signs [117]; however, the effect of these treatments on corneal opacity with MPS remains unclear and likely specific to the particular disorder, disease severity, and the degree of donor chimerism [118,119]. Corneal transplantation may be beneficial for some patients with significant corneal opacity [120]. Additionally, AAV8G9-opt-IDUA (AAV8 and 9 chimeric capsid-optimized-*IDUA*) gene therapy shows great promise to prevent and potentially reverse MPS I-associated corneal blindness [45,49,55]. Likewise, adenoviral-mediated expression of β-glucuronidase (GUSB) transgene in the stromal region has been shown to reduce corneal clouding in MPS VII mouse models, and similar results were reported in canine MPS VII models following adenovirus-mediated transgene expression of human GUSB [110,111]. AAV2/8-mediated transgene expression of ARSB (arylsulfatase B) through intravenous administration showed efficient liver transduction and favorable outcomes in the liver, heart, and bone of MPS VI feline models [112]; however, outcome in the cornea was not evaluated in these studies. It must be noted that due to the relative avascular nature of the cornea, a systemic rAAV delivery approach might not efficiently transduce the cornea; therefore, a localized AAV-mediated ARSB approach could be potentially therapeutic. This local AAV vector administration to the cornea likely can complement the defects in treating MPS diseases by bone marrow transplantation and perhaps be used in conjunction with an AAV vector approach to treat other disease symptoms (either local or systemic) [114,119]. Of particular note is that redosing of AAV vectors has been observed in mice and administration of AAV vectors to the rabbit cornea resulted in minimal to no serum capsid neutralizing antibodies [46,55]. These combined results suggest that AAV vector treatment of lysosomal storage diseases can be used in combination with AAV vectors to treat other disease aspects outside the cornea.

### 6.7. Corneal Graft Rejection

Corneal engraftment to treat vision loss is the most common form of tissue transplantation worldwide with approximately 47,000 transplants occurring annually in the United States alone. Corneal transplantation, such as penetrating keratoplasty and Descemet stripping endothelial keratoplasty (DSEK), is a common treatment for endothelial abnormalities (i.e., Fuchs endothelial dystrophy) and keratoconus [115]. Corneal grafts are considered at high risk for rejection when there is a loss of any component of corneal immune privilege [65], which may occur with ocular inflammation, infection, trauma, and subsequent vascularization. In low risk patients, transplant rejection after 2 years is approximately 15%, the success of which is largely due to the lack of vascularization in relatively healthy corneas. In contrast, high risk corneal transplants demonstrate an alarming 50–70% rejection rate after only 2 years, and many patients with severe corneal disease are not considered good candidates for the procedure [115]. These high-risk cases are defined as having a significant amount of pre-existing corneal vascularization and/or having had a prior engraftment. Corneal graft rejection develops within weeks after surgery and is manifested clinically by an epithelial or endothelial rejection line, stromal rejection band, increased corneal thickness, and anterior segment inflammation (keratic precipitates, aqueous flare, and cells) [65]. Current treatments, in part, rely on topical and systemic corticosteroids that exhibit low levels of success and serious adverse side effects. Gene therapy can be implemented to reduce immune mediated corneal graft rejection through reducing corneal neovascularization, dampening the immune response to the cornea, and inhibiting apoptotic cell death of the cornea. Strategies to reduce corneal neovascularization are discussed in Section 6.8. Towards dampening the immune response of the cornea, ex vivo lentiviral-mediated delivery of indoleamine 2,3 dioxygenase, an immunomodulatory enzyme, significantly prolonged corneal graft survival in mice. Yet another ex vivo lentiviral-mediated delivery strategy employing Bcl-xl, mammalian anti-apoptotic protein, and p35, a baculoviral anti-apoptotic protein, lead to significant enhancement of graft survival [121]. AAV-gene therapy to deliver HLA-G (serotype 8G9) [73], an immunomodulatory and anti-inflammatory molecule, or IL-10 (serotype 2) [122], an anti-inflammatory cytokine, has also shown promise for improving corneal graft survival.

### 6.8. Corneal Neovascularization

The avascular structure of the cornea is essential for the preservation of optimal vision. Neovascularization describes the condition of abnormal growth of new capillaries towards the cornea from the limbus region usually due to injury or infection. Neovascularization has been shown to occur in a wide variety of corneal disorders, including, mechanical injury, chemical burns, infectious keratitis, corneal graft rejection, and insults from lens wear. Such blood vessels may lead to persistent inflammation, scarring, and vision obstruction. Likewise, excessive corneal neovascularization contributes to corneal graft rejection. Corneal neovascularization is estimated to develop in 1.4 million patients per year in the United States [123]. Currently drugs used to control corneal neovascularization include anti-inflammatory drugs (topical steroids and/or NSAIDs), anti-VEGF agents (topical and/or subconjunctival administration of bevacizumab or ranibizumab), and/or MMP inhibitors (oral doxorubicin combined with topical corticosteroids), while surgical options include laser photocoagulation. These treatments often display partial efficacy and may result in additional ocular complications [124]. Therefore, gene therapy-mediated treatments against corneal neovascularization may be beneficial in patients.

VEGF has been reported to be essential for neovascularization in the cornea; therefore, gene therapy approaches to inhibit VEGF signaling have been studied as a potential therapy. Subconjunctival injection of AAV2-endostatin, a potent anti-angiogenesis agent, showed sustained transgene expression with a minimal immune response, followed by inhibition of neovascularization in corneal neovascularization mouse models [5]. Similarly, targeted stromal delivery of AAV5-decorin, an angiogenesis regulator, through the removal of corneal epithelium and topical application onto the stroma for two minutes showed a significant reduction of corneal neovascularization in a VEGF-induced rabbit neovascularization model [70]. Intravitreal injection of Ad-mediated sFlt-1, (an antiangiogenic factor) showed reduced corneal angiogenesis in silver nitrate/potassium nitrate corneal cauterization rat models [125]. AAV-mediated delivery of sFlt1 may be a potential avenue to investigate as treatment of corneal neovascularization. Additionally, introducing immunomodulatory molecules through gene therapy has been pursued as a possible strategy to mitigate corneal neovascularization. Towards this effort, transduction of HLA-G, an immunomodulatory and anti-vascularization molecule, in rabbit corneas were shown to prevent burn-induced corneal neovascularization [73]. Lastly, Lu et al. performed corneal miRNA (microRNA) profiling following alkali-burn treatment in the mouse to find 36 highly upregulated and 3 strongly downregulated miRNAs compared to non-treated controls. miR-204, one of the downregulated miRNA candidates, was subsequently evaluated as a potential gene therapy candidate for inhibiting angiogenesis in the cornea. rAAVrh.10 mediated delivery of miR-204 showed reduced vascularization of injured mouse cornea [126]. AAV gene therapy has shown efficacy and safety in several animal models; therefore, subsequent clinical and preclinical studies should allow for the development of new treatments for patients with corneal neovascularization.

## 7. Route of Administration for AAV Gene Therapy

A key step in enhancing targeted efficacy and reducing toxicity from gene therapy is through selective gene delivery into the appropriate cells. The location of the cornea allows for more accessibility for gene delivery. Currently, topical application, intrastromal, subconjunctival, and intracameral injections have been reported with outcomes summarized in Table 2 (Figure 3, Table 2). Due to the nature of the eye, a topical administration presents a simple route of administration. However, simple drop application of AAV vectors have demonstrated relatively low levels of transduction without the removal superficial epithelium, while also presenting concerns of AAV transduction of non-target individual due to viral shedding through tears. Intrastromal injections of AAV serotypes 1, 2, 5, and 8 in mouse cornea showed preferential transduction of the stroma, whereas intrastromal injection of AAV serotype AAV8G9 in dog, rabbit, and human ex vivo eye models transduce the entire cornea (Table 2). Intracameral injection of different AAV serotypes in different animal models has shown preferential transduction of corneal endothelium (Table 2). Although these injection routes show effective corneal transduction, transient corneal damage is often associated with these injection routes; therefore, they are not commonly used in the clinic. In contrast, subconjunctival injections are commonly used for other optical applications; however, subconjunctival injection of AAV was poorly defined. Recently, Song et al. performed subconjunctival injections of AAV serotypes 2, 6, and 8 in C57BL/6J mice to show serotype-specific transduction of GFP in different corneal tissues. The study showed preferential expression of GFP in the stroma with AAV8, in the endothelium with AAV6, while GFP expression was not detectable in the cornea of AAV2-injected mice [127].

## 8. Immune Response Following rAAV Gene Therapy

The immune system has evolved to mitigate the deleterious effects of foreign infections and damaged cells in the body. Immune responses are also mounted against AAV gene therapy, which can ultimately dampen the intended efficacy of the original therapy and prevent a patient from receiving a second round of AAV-mediated therapy. In the therapeutic context, the immune response towards AAV vectors occurs at least in three levels: (1) an antibody response to the AAV capsid, which can be pre-existing in patients due to prior infection from wild-type AAV or induced following AAV vector administration, (2) a T-cell response to a foreign transgene product, and (3) an innate immune response to the rAAV transduction. Although only a limited number of reports using AAV gene therapy in the cornea have been published to date, the physiology of this unique organ positions itself as an ideal organ for gene therapy with reduced immunological concerns.

Features such as an avascular surface or lack of lymphatic drainage from the anterior chamber make the cornea a relatively immune privileged tissue in the body. Moreover, ACAID induces additional immune-tolerance to foreign antigens injected into the eye’s anterior chamber (reviewed in [139]). Compartmentalization of the eye in general and the use of a restricted intrastromal delivery route allows for low therapeutic AAV vector doses and minimal to no peripheral organ (or systemic) vector genome distribution. For instance, AAV vectors were not found outside the ocular compartment following intrastromal injection of AAV-opt-IDUA in a WT rabbit model [55]. Additionally, even at a relatively high vector dose, WT rabbits administered rAAV completely tolerated the production of a human protein in the corneal stroma with no signs of adverse reactions attributed to a deleterious innate immune response [55]. In the same context, 6 months post-injection to the corneal stroma, only a single animal demonstrated a weak capsid neutralizing serum antibody response, with no neutralizing antibodies found in the aqueous or vitreous humor [55]. However, transient corneal edema was noted following AAV vector injection in MPS I canines [49], which unlike human MPS I patients or WT rabbits display corneal neovascularization, which compromises for ocular immune privilege. Although, this corneal edema was theoreticized to be a T-cell response to the human *IDUA* protein in MPS I canines, several months later at the timepoint of humane euthanasia, strong *IDUA* protein was observed by several measures [49]. Collectively, precise injections of AAV vectors the corneal stroma are well tolerated and result in minimal to no vector dissemination, no T cell responses to foreign protein production, and in general the lack of antibody response to the AAV capsid. However, if vector is administered to a cornea that is compromised for immune privilege then the potential for a negative immune response and vector dissemination increases [49,73]. In general, further research in diverse disease models is required to better understand the immune response to AAV corneal gene therapy. Minimal overall dose (compared to a systemic injection route) required for corneal transduction, thereby reducing the subsequent immune response to the AAV gene therapy.

## 9. Unanswered Questions in AAV Corneal Gene Therapy

AAV corneal gene therapy appears to have many favorable attributes compared to most routes of vector administration, however, there remain unanswered questions including potential safety concerns and overall durability of the treatment effect/transgene persistence. For instance, AAV vector genome shedding has been observed in tears for up to two weeks following intrastromal injection, however intact vector particles that elicit transduction was not thoroughly investigated [49]. Although vector shedding is observed following most administration routes of AAV gene therapy for months to years, it may be wise to use precautions for family member and caregivers following corneal gene therapy. This concern could be amplified by the potential for AAV vector mobilization as WTAAV and helper viruses such as adenovirus and herpes simplex virus often infect the human cornea [140,141]. Another limitation of AAV vectors, in general, and for corneal gene therapy is the possibility for vector genome integration into the host chromosome, especially in dividing cells [142]. Work in MPS I canines administered AAV vectors demonstrated basal epithelia transgene positive cells at a surprising level several months after corneal injection [49]. Depending on the particular disease, this could be beneficial or perhaps results in corneal intraepithelial neoplasia, which has yet to be observed in any corneal context. The durability of AAV vector expression following corneal administration also remains to be defined. For instance, corneal keratocytes are normally quiescent with minimal turnover; however, trauma and/or disease states induce differentiation and/or cell death, which may influence the capacity for long-term transgene expression following a single injection [143]. Although, experiments to date in mice, canines, and rabbits have demonstrated the persistence of the transgene derived product, and transgenic genomes, out to 17, 6, and 6 months following a single administration, respectively [46,49,55]. Finally, the immune response, whether humoral or cell mediated, to AAV gene therapy requires further elucidation. Of particular note, was a transient single appearance of corneal edema observed in a subset of MPS I canines administered AAV vectors encoding foreign transgenes to the corneal stroma ranging from 5 to 19 weeks post-injection [49]. The observed edema was not characterized and resolved following steroid treatment. The kinetics of the appearance of the corneal edema suggests a response to the transgene product based on observations observed to AAV-derived proteins that are foreign to the recipient, however, the abundant transgene product was observed in these corneas at the experimental conclusion [49,56]. To our knowledge, this observation of a potential corneal immune response to AAV vectors in MPS I canine corneas is the only instance in which one has been suggested and the authors hypothesize it may be specific the MPS I canine model in which ocular immune privilege is compromised due to corneal neovascularization. Collectively, although corneal gene therapy may have several of the concerns associated with almost all other routes of AAV vector administration, as it has historically been relatively understudied, additional studies are required to better elucidate the widespread application of AAV vectors for corneal gene therapy.

## 10. Challenges and Conclusions

For corneal diseases, advances in detection tools, anti-inflammatory therapies, pain-management therapies, and corneal surgery including corneal transplantations, have improved clinical outcomes; moreover, with improved understanding of disease pathology, future studies should be focused on developing better therapies. AAV-mediated gene therapy targeting the cornea has several advantages compared to applications in other areas of the body. The anterior eye is relatively accessible, allowing AAV vectors to be administered by direct injection or a topical application to the ocular surface. Furthermore, the relatively small tissue size of this compartment and the relatively efficient transduction capability of AAV allows for the use of a significantly smaller overall volume and dose compared to systemic applications [49,55]. These attributes, coupled with at least partial restriction to the ocular compartment minimizes systemic drug dissemination [55], thus minimizing the immunologic response or toxicity to the therapy. Additionally, the lengthy viability of human corneas post-mortem offers the availability of relevant tissue for optimization experiments, at least when targeting the cornea, which minimizes interspecies translation concerns regarding vector tropism. Further work is required to determine the most effective AAV serotype for each route and specific cell type transduced, which is important for the success of potential therapies. Although AAV vectors show broadly efficient transduction in the cornea, tailoring the serotype and administration route for a specific application will depend on the disease. Furthermore, concerns of an unwanted immune response, systemic biodistribution with off-target expression, and AAV vector shedding and dissemination to non-patients should be considered when assessing the potential therapeutic benefits. Nonetheless, given the safety and tempered successes of AAV vectors in the posterior eye, both research and patient communities remain optimistic that AAV gene therapy can offer vision restoration to the millions of people suffering from corneal diseases.

## Figures and Tables

**Figure 1 pharmaceutics-12-00767-f001:**
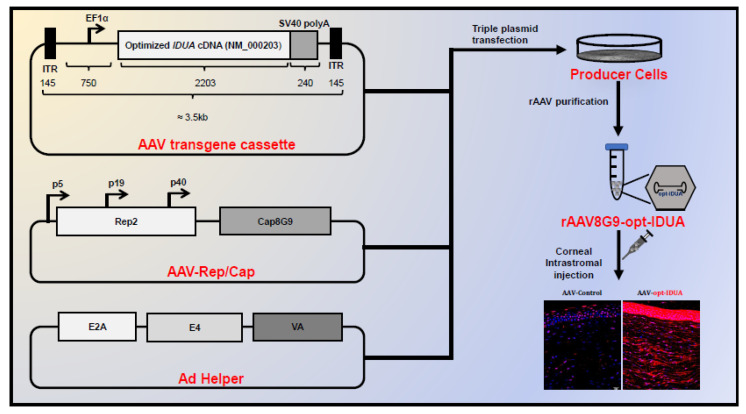
Outline of the AAV vector design, production, and transduction of the canine cornea. Triple transfection protocol is utilized to replicate and package the optimized *IDUA* expression cassette harboring EF1α promoter and SV40 polyA flanked by ITR inside *cap8G9* using Hek293 producer cells. After purification AAV8G9-opt-IDUA virions (or AAV8G9-GFP control vectors) are delivered to the cornea by intrastromal injection in a canine model. Immunofluorescence staining for *IDUA* (red) was performed on histological sections 8 months post-injection and on-target expression of recombinant *IDUA* is observed in corneal keratocytes.

**Figure 2 pharmaceutics-12-00767-f002:**
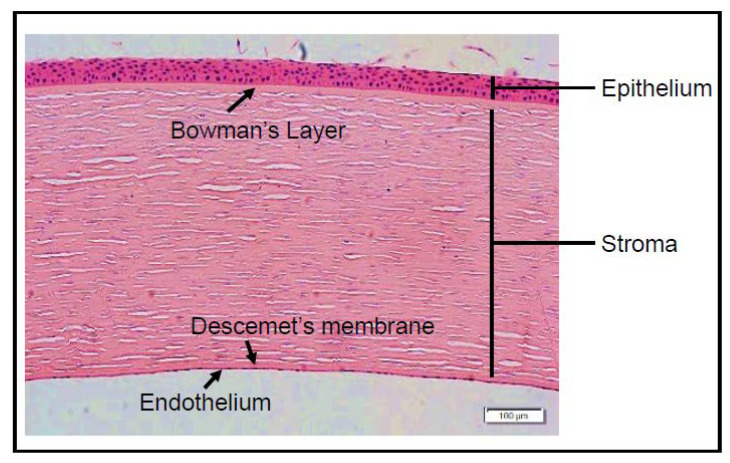
H&E image of a human cornea. A representative image of a human cornea. Corneal layers from anterior to posterior include epithelium, Bowman’s layer, stroma, Descemet’s membrane, and endothelium, respectively. The scale represents 100 µm in length.

**Figure 3 pharmaceutics-12-00767-f003:**
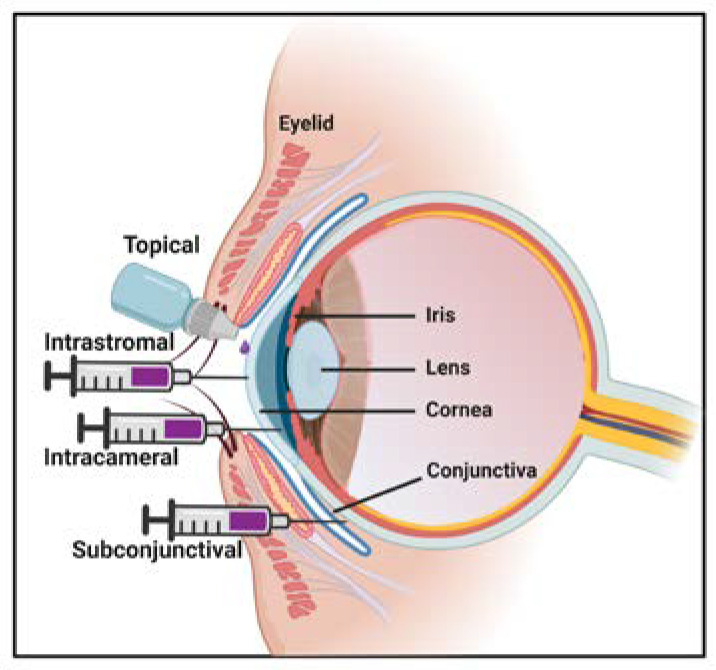
Injection routes for corneal gene therapy. Reported routes for gene therapy targeting the cornea have included direct intrastromal injection, intracameral injection, topical application (incubation after epithelial removal), and subconjunctival injection.

**Table 1 pharmaceutics-12-00767-t001:** Common corneal dystrophies along with associated genes and pattern of inheritance.

Disorders	Genes/Chr Location	Inheritance Pattern
CD, Avellino Type	*TGFBI*	autosomal dominant
CD, Congenital Endothelial 1	20p11.2–q11.2 locus	autosomal dominant
CD, Congenital Stromal	*DCN*	autosomal dominant
CD, Epithelial Basement Membrane	*TGFBI*	autosomal dominant
CD, Fleck	*PIKFYVE*	autosomal dominant
CD, Fuchs Endothelial, Early Onset	*COL8A2*	autosomal dominant
CD, Fuchs Endothelial, Late Onset	*ZEB1*	autosomal dominant
CD, Fuchs Endothelial, Late Onset 2	*TCF4*	autosomal dominant
CD, Granular	*TGFBI*	autosomal dominant
CD, Lattice Type I	*TGFBI*	autosomal dominant
CD, Lattice Type II	*GSN*	autosomal dominant
CD, Meesmann	*KRT12*, *KRT3*	autosomal dominant
CD, Posterior Amorphous	12q21.33 deletion	autosomal dominant
CD, Posterior Polymorphous 1	*OVOL2*	autosomal dominant
CD, Posterior Polymorphous 2	*COL8A2*	autosomal dominant
CD, Posterior Polymorphous 3	*ZEB1*	autosomal dominant
CD, Posterior Polymorphous 4	*GRHL2*	autosomal dominant
CD, Recurrent Epithelial Erosions	unknown	autosomal dominant
CD, Reis-Bücklers	*TGFBI*	autosomal dominant
CD, Schnyder	*UBIAD1*	autosomal dominant
CD, Stocker-Holt	*KRT12*	autosomal dominant
CD, Subepithelial Mucinous	unknown	autosomal dominant
CD, Thiel-Behnke	*TGFBI*	autosomal dominant
CD, Band-Shaped	unknown	unknown
CD, Congenital Endothelial 2	*SLC4A11*	autosomal recessive
CD, Gelatinous Drop-like	*M1S1 (TACSTD2)*	autosomal recessive
CD, Macular	*CHST6*	autosomal recessive
CD, Lisch Epithelial	unknown	X-linked dominant
CD, Endothelial X-Linked	Xq25 locus	X-linked unclear*

Notes: CD, corneal dystrophy; unclear*, both inheritance pattern (dominant and recessive) reported in the literature; source: a database of hereditary ocular disease, hereditary ocular disease, The University of Arizona Health Sciences.

**Table 2 pharmaceutics-12-00767-t002:** Cornea route of administration and transduced cells/section.

Transduced Cell/Section	Route of Administration	Species (Model)	Dose/Volume	Serotype	Promoter	Transgene	Ref
Stroma	Intrastromal	C57BL/6 mouse	1 × 10^9^ vg/2 µL	AAV1, 2, 5, 8	CMV	EGFP	[46]
Stroma	Intrastromal	Human (ex vivo)	5 × 10^10^ vg/300 µL	AAV1, 2, 8	CMV	EGFP	[46]
Entire cornea	Intrastromal	Human (ex vivo)	1 × 10^10^ vg/50 µL	AAV8G9	CMV	GFP, IDUA	[45]
Entire cornea	Intrastromal	MPS I dogs	5–8 × 10^10^ vg/50–80 µL	AAV8G9	CMV	GFP, IDUA	[49]
Entire cornea	Intrastromal	Male New Zealand white rabbit	5 × 10^10^ vg/50 µL	scAAV8G9	JET	GFP, HLA-G	[73]
Stroma & Epithelium	Intrastromal	C57BL/6J mouse	1.4 × 10^11^ vg/2 µL	AAV2/5, 2/8, 2/9, 2/8Y733F, AAV[ShH10], AAV[Anc80]	CMV	EGFP	[128]
Entire cornea	Intraperitoneal	Ai9 mouse	7.2 × 10^11^ vg/10 µL	AAV9	CMV	Cre	[128]
Endothelium	Intracameral	New Zealand white rabbits	1 × 10^7^ vg/25 µL	AAV2	CMV	LacZ	[129]
Endothelium	Intracameral	Brown Norway & Wistar Rat	3 × 10^9^–6 × 10^9^ VP/3–5 µL	ssAAV2	CMV	GFP	[130]
Endothelium	Intracameral	Cynomolgus monkey	3 × 10^10^ VP/30 µL	scAAV2	CMV	GFP	[130]
Endothelium	Intracameral	C57BL/6 mouse	9 × 10^8^ vg–3 × 10^9^ vg/1 µL	scAAV2, scAAV2 (variants), scAAV8(variants)	CMV-CBA	GFP	[131]
Endothelium	Intracameral	Sprague Dawley rat	1.8 × 10^9^ vg–6 × 10^9^ vg/2 µL	scAAV2, scAAV2 (variants), scAAV8 (variants)	CMV-CBA	GFP	[131]
Endothelium	Intracameral	C57BL6 mouse	4 × 10^9^ vg/2 µL or 1 × 10^11^ vg/2 µL	AAV9	CMV, Tet	EGFP, MMP-3	[132]
Epithelium	Topical ^a^	Rat	Unknown/20 µL	AAV2	CAG	EGFP	[133]
Keratocyte	Topical ^b^	New Zealand white rabbit	5 × 10^11^ VP/25 µL or 1 × 10^11^ VP/10 µL	Not specified	CMV	β-gal, CAT	[134]
Entire cornea	Topical ^c^	New Zealand white rabbit	2 × 10^11^ vg	AAV1, 2, 5, 7, 8	CMV-CBA	GFP	[135]
Entire cornea	Topical ^c^	Human cornea (ex vivo)	1.2–7.8 × 10^10^ vg	AAV1, 2, 5, 7, 8	CMV-CBA	GFP	[135]
Keratocytes	Topical ^d^ or microinjection	Mouse	Not specified	AAV2, 5	CMV-CBA	EGFP	[136]
Keratocytes	Topical ^e^	C57 Mouse	2 × 10^9^ vg/2 µL	AAV6, 8, 9	RSV	Alkaline phosphatase	[47]
Stroma	Topical ^f^	Female C57 black mouse	2.2 × 10^8^ vg/2 µL	AAV8	RSV	Alkaline phosphatase	[137]
Stroma	Topical ^f^	New Zealand white rabbit	5 × 10^11^ vg/100 µL	AAV5	CMV-CBA	GFP, Decorin	[70]
Keratocyte	Topical ^g^	New Zealand white rabbit	6.5 × 10^11^ vg/100 µL	AAV5	CMV-CBA	GFP	[138]
Stroma	Topical ^h^	New Zealand white Rabbit	2.0 × 10^12^/75 µL	AAV5	CMV-CBA	Smad7	[71]
Epithelium	Subconjunctival	CD-1 mouse	2.5 × 10^7^ VP/5 µL	AAV2	CMV	EGFP, Endostatin	[5]
Endothelium	Subconjunctival	C57BL/6J mouse	7 × 10^9^ vg/14 µL	AAV6	CMV	GFP	[127]
Stroma	Subconjunctival	C57BL/6J mouse	7 × 10^9^ vg/14 µL	AAV8	CMV	GFP	[127]

Notes: ^a^: Topical application after removal of superficial epithelium; ^b^: Topical application beneath a lamellar flap; ^c^: Topical application with a 10 mm diameter trephine after removal of a depth of 25 μm of epithelium by excimer laser; ^d^: Topical application on de-epithelialized cornea; ^e^: Topical application on de-epithelialized cornea after drying with merocel sponge; ^f^: Topical application on de-epithelialized cornea using a custom-cloning cylinder; ^g^: Topical application on de-epithelialized cornea after drying with hairdryer; ^h^: Topical application after keratectomy; scAAV2 (septuple): scAAV2(Y252+272+444+500+700+704+730F); TetOn: Tetracycline-inducible promoter; CMV-CBA: cytomegalovirus enhancer chicken β-actin promoter; Y-F mutation: tyrosine to phenylalanine mutations; DPI: Days post injection; WPI: weeks post injection; CNV: Corneal neovascularized; vg: vector genome; vp: viral particle; IU: infection unites; PDGF-B: human platelet-derived growth factor B; β-NGF: human β-nerve growth factor.

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
