# Peer review of "Adeno-Associated Virus Mediated Gene Therapy for Corneal Diseases"

_pharmaceutics, 2020, doi:10.3390/pharmaceutics12080767_

Round 1

Reviewer 1 Report

In this review manuscript, Bastola and co-workers provide an extensive and detailed overview of the gene therapy strategies than leverage adeno-associated virus (AAV) for corneal diseases. The authors introduce briefly gene therapy and AAV biology as well as the pioneering efforts to develop AAV-based vectors as a gene transfer tool. The authors then describe the corneal structure and associated disorders. For each disorder, the authors describe how AAVs may be exploited as a therapeutic tool. The authors conclude the manuscript describing the routes of AVV administration and challenges of AAV employment such as its immunogenicity.     

The topic is of interest and the references seem up to date. The paper is well written. I could not find any major deficits in the manuscript.

Chapter 4 ‘Targeting disease with AAV gene therapy’ is partially repeated in chapter 6.6 ‘corneal opacity associated with mucopolysaccharidoses’ and in the introduction. Therefore chapter 4 could be integrated in chapter 6.6, and it is not necessary to repeat the introduction.

Chapter 5 ‘Cornea: Structure and function’ has some repetitions inside the paragraph (e.g.  acellular layers are described twice).

Author Response

Reviewer #1

In this review manuscript, …………………………

Chapter 4 ‘Targeting disease with AAV gene therapy’ is partially repeated in chapter 6.6 ‘corneal opacity associated with mucopolysaccharidoses’ and in the introduction. Therefore chapter 4 could be integrated in chapter 6.6, and it is not necessary to repeat the introduction.

Our response:

  • We feel chapter 4 provides a basic template to utilize AAV towards targeting human disorders; therefore, we think that the information provided in this section is of interest to the AAV community in general. Having said that, we agree that some parts of this sections may seem redundant; therefore, we have decided to remove some sentences from chapter 4 that were also included in the introduction.

Chapter 5 ‘Cornea: Structure and function’ has some repetitions inside the paragraph (e.g.  acellular layers are described twice).

Our response:

  • We agree with the reviewer that some descriptions of the cornea seem redundant; therefore, according to the reviewer’s suggestion, we have removed the initial description of cellular and acellular components of the cornea from the first paragraph of chapter 5 (as it is also described in more detailed later in Chapter 5).

Reviewer 2 Report

This manuscript provides a review on AAV vectors and their use for gene therapy for corneal diseases. Overall, this is a well-organized review covering basic AAV biology, its use as a gene therapy vector, and its potential utility for both genetic and non-genetic corneal diseases. The authors have done rigorous reviews on the literature in the relevant fields. The manuscript cites appropriate and helpful references including review papers and cutting-edge studies so that it benefits broader readership with varying interests in gene therapy and ocular diseases. The following comments are provided for the authors to further improve this manuscript.

Major points:

The introduction and background on AAV and AAV vectors is quite long, using >200 lines out of a total of 723 lines. This has made the review somewhat unfocused. There already exist excellent review papers on AAV vectors and AAV vector-mediated gene therapy. In light of the main theme of this review, that is, AAV vector-mediated corneal diseases, it is most likely that readers would be better served by reading separate reviews papers on AAV vectors (those cited in this manuscript) followed by a focused review on just corneal gene therapy provided in by this manuscript.

The review provides a very optimistic outlook for AAV-mediated corneal gene therapy. The authors highlight the limitations of standard care and the potential for AAV-mediated gene therapy. Sometimes, the rationale for gene therapy appears to simply be that is not the current standard of care (for example, see lines 593-595). The authors do not mention specific limitations and obstacles in the field, but make broad statements regarding the limitations of AAV-mediated gene therapy in general (at the end). What are the major bottlenecks to achieving successful gene therapy in each major corneal disease class? Are the cited pre-clinical studies ripe for translation, or does more work need to be done? More balance would be a welcome addition to this otherwise informative review.

Minor points:

Errors are found in quite a few places. Few examples among many are given as follows. Please carefully edit the manuscript before re-submission. Words in parentheses are suggested corrections.  Line 417 "United State" (United States); Line 527 "AAV platform" (An AAV platform or AAV platforms); Line 532 "Therefore, AAV mediated" (Therefore, an AAV-mediated); Lines 527-530 - some words appear to be missing after the second comma; Lines 34 and 550 - mHTT and ARSB need to be spelled out. Line 657 "completed" (completely); and Line 670 “than” (then).

Please clearly describe AAV serotypes in the main text that were used in the studies cited in the paper.

Line 84. "AAV genome includes one poly-A tail" is not correct.

Line 162. What are the 191 nucleotide AAV ITRs? AAV2 ITR is 145-nt long and the commonly used AAV2 vector plasmids contain 130-bp ITRs.

Lines 436-438. This appears to be an entirely hypothetical statement. Do the authors have compelling scientific premise for this statement?

Lines 491-505. HDR in quiescent cells is likely inefficient. How do the authors apply CRISPR HDR approaches to corneal diseases? Successful in vivo approaches in this context use growing hepatocytes (neonatal injection).

Line 552. Some clarifications are needed. Please clarify whether the authors would expect a systemic treatment to efficiently reach the cornea and whether they propose a localized treatment as a supplementary systemic gene therapy for MPS IV.

Lines 675-703. This section "9. Corneal diseases with unmet need" appears redundant. Could it be made more concise and added into the next (summary) section?

Author Response

Reviewer #2

This review manuscript, …………………………

The introduction and background on AAV and AAV vectors is quite long, using >200 lines out of a total of 723 lines. This has made the review somewhat unfocused. There already exist excellent review papers on AAV vectors and AAV vector-mediated gene therapy. In light of the main theme of this review, that is, AAV vector-mediated corneal diseases, it is most likely that readers would be better served by reading separate reviews papers on AAV vectors (those cited in this manuscript) followed by a focused review on just corneal gene therapy provided in by this manuscript.

Our response:

  • We appreciate the reviewer’s comment regarding the length of the introduction and background on AAV and AAV vectors. While we attempted to present a summarized version of the AAV reviews that were cited in this manuscript, we agree with the reviewer that some areas of our manuscript might be considered to be lengthy. We, therefore have deleted some sentences which has now reduced ~22 lines from the AAV background and AAV vector section (previously up to line 245, now up to line 223). We hope that the editor and the reviewer will find the current version to be more focused.

The review provides a very optimistic outlook for AAV-mediated corneal gene therapy. The authors highlight the limitations of standard care and the potential for AAV-mediated gene therapy. Sometimes, the rationale for gene therapy appears to simply be that is not the current standard of care (for example, see lines 593-595). The authors do not mention specific limitations and obstacles in the field, but make broad statements regarding the limitations of AAV-mediated gene therapy in general (at the end). What are the major bottlenecks to achieving successful gene therapy in each major corneal disease class? Are the cited pre-clinical studies ripe for translation, or does more work need to be done? More balance would be a welcome addition to this otherwise informative review.

Our response:

  • We thank the reviewer for raising these very important questions and agree more balance was needed. We therefore have included a new section “Section 9: Unanswered questions in AAV corneal gene therapy (Line 662- 697)”, where we tried to answer the questions raised by the reviewer. We hope that reviewer and the editor will find our answers in line with the pertinent questions raised by the reviewer.

Reviewer’s Minor points:

Errors are found in quite a few places. Few examples among many are given as follows. Please carefully edit the manuscript before re-submission. Words in parentheses are suggested corrections.

 Line 417 "United State" (United States);

Our response:

  • We have corrected this error (Now Line 391)

Line 527 "AAV platform" (An AAV platform or AAV platforms);

Our response:

  • We have corrected this error (Now Line 503)

Line 532 "Therefore, AAV mediated" (Therefore, an AAV-mediated);

Our response:

  • We have corrected this error (now Line 508)

Lines 527-530 - some words appear to be missing after the second comma;

Our response:

  • We-wrote the sentence (now Line 503-506)

Lines 34 and 550 - mHTT and ARSB need to be spelled out.

Our response:

  • We spelled out mHTT (now line 34) and ARSB (now line 528) as suggested by the reviewer.

Line 657 "completed" (completely); and Line 670 “than” (then).

Our response:

  • We corrected both errors as suggested by the reviewer (Now Lines 644 and 657, respectively).

Please clearly describe AAV serotypes in the main text that were used in the studies cited in the paper.

Our response:

  • We have included AAV serotype information in the main text as suggested by the reviewer.

Line 84. "AAV genome includes one poly-A tail" is not correct.

Our response:

  • While the specifics of AAV transcript termination may be underestimated, and perhaps serotype independent, in our statement we were following data from the well cited report from Dr. Bern’s lab (PMID: 6300419). We have edited, and now cited, the statement as follows: “AAV2 genome primarily includes two main protein coding regions and one poly A tail, flanked at both ends by 145 nucleotides inverted terminal repeats (ITRs).

Line 162. What are the 191 nucleotide AAV ITRs? AAV2 ITR is 145-nt long and the commonly used AAV2 vector plasmids contain 130-bp ITRs.

Our response:

  • The study described included the 191 nucleotides had incorporated the AAV2-ITRs with some additional sequences. The sentence has been re-written to reflect this fact (Now Line 144).

Lines 436-438. This appears to be an entirely hypothetical statement. Do the authors have compelling scientific premise for this statement?

Our response:

  • We thank the review for pointing this out and agree that the statement might seem hypothetical. Therefore, we included some additional information to our statement suggesting that since anti-inflammatory medications such as cyclosporine and lifitegrast are already used to manage DED; therefore, an AAV gene therapy could be utilized to modulate the immune response. We hope that this will strengthen our suggestion to use AAV gene therapy to target the immune response. (Now Line 409-411)

Lines 491-505. HDR in quiescent cells is likely inefficient. How do the authors apply CRISPR HDR approaches to corneal diseases? Successful in vivo approaches in this context use growing hepatocytes (neonatal injection).

Our response:

  • We agree and thank the reviewer for asking this very important question. The study described in this statement was conducted in vitro and the authors have yet to evaluate the CRISPR HDR efficacy in vivo. As such, we have modified the sentence, to include this information.

Taketani et al. showed correction of R124H mutation in TGFBI, which causes granular corneal dystrophy (GCD) in primary human corneal keratocytes in vitro; however, HDR efficacy of corneal keratocytes in vivo were not evaluated by this group”. (Now Line 474-475)

Some clarifications are needed. Please clarify whether the authors would expect a systemic treatment to efficiently reach the cornea and whether they propose a localized treatment as a supplementary systemic gene therapy for MPS IV.

Our response:

  • We thank the reviewer for pointing this out. Based on the reviewer’s suggestion, we have included more information regarding the efficacy of systemic AAV transduction to the cornea. Currently the sentence reads as follows:

It must be noted that due to the relative avascular nature of the cornea, a systemic rAAV delivery approach might not efficiently transduce the cornea; therefore, a localized AAV-mediated ARSB approach could be potentially therapeutic …..” (Now Line 530-539)

Lines 675-703. This section "9. Corneal diseases with unmet need" appears redundant. Could it be made more concise and added into the next (summary) section?

Our response:

  • We agree with the reviewer that section 9 “Corneal diseases with unmet need” may appear redundant. While drafting this section (and as the section title suggest), we were attempting to concisely present corneal diseases that currently would benefit from AAV gene therapy; however, we see the reviewer’s point and have decided to remove this section.